



# Experimental evidence for a universal threshold characterizing wave-induced sea ice break-up

Joey Voermans[1], Jean Rabault[2,3], Kirill Filchuk[4], Ivan Ryzhov[4], Petra Heil[5], Aleksey Marchenko[6], Clarence Collins[7], Mohammed Dabboor[8], Graig Sutherland[9], and Alexander Babanin[1,10]

[1]Department of Infrastructure Engineering, University of Melbourne, Parkville, Australia
[2]Norwegian Meteorological Institute, Oslo, Norway
[3]Department of Mathematics, University of Oslo, Oslo, Norway
[4]Arctic and Antarctic Research Institute (AARI), St. Petersburg, Russian Federation
[5]Australian Antarctic Division and Australian Antarctic Program Partnership, University of Tasmania, Hobart, Australia
[6]The University Centre in Svalbard, Longyearbyen, Norway
[7]Coastal and Hydraulics Laboratory, U.S. Army Corps of Engineers, Kitty Hawk, USA
[8]Science and Technology Branch, Environment and Climate Change Canada, Dorval, Canada
[9]Environmental Numerical Prediction Research, Environment and Climate Change Canada, Dorval, Canada
[10]Laboratory for Regional Oceanography and Numerical Modeling, National Laboratory for Marine Science and Technology, Qingdao, China

**Correspondence:** Joey Voermans (jvoermans@unimelb.edu.au)

**Abstract.** Waves can drastically transform a sea ice cover by inducing break-up over vast distances in the course of a few hours. However, relatively few detailed studies have described this phenomenon in a quantitative manner, and the process of sea ice break-up by waves needs to be further parameterized and verified before it can be reliably included in forecasting models. In the present work, we discuss sea ice break-up parameterization and demonstrate the existence of an observational

threshold separating breaking and non-breaking cases. This threshold is based on information from two recent field campaigns, supplemented with existing observations of sea ice break-up. The data used cover a wide range of scales, from laboratory-grown sea ice to polar field observations. Remarkably, we show that both field and laboratory observations tend to converge to a single quantitative threshold at which the wave-induced sea ice break-up takes place, which opens a promising avenue for robust parametrization in operational forecasting models.

## 1 Introduction

Surface gravity waves can propagate tens to hundreds of kilometers into the ice pack before the ice fully dissipates their energy (e.g., Kohout et al., 2014; Stopa et al., 2018). In the process, waves flex the ice, imposing stresses on the elastic and brittle ice sheet. When these stresses exceed a critical value, the sea ice will crack or break, creating large regions of broken ice floes

with complex dynamics (Horvat et al., 2016; Hwang et al., 2017). Once broken, the ice is able to move more freely, reducing





the attenuation of wave energy (e.g., Collins et al., 2015), and thereby allowing waves to penetrate even further into the ice pack. This drives a series of secondary processes in the coupled air-sea system that can further affect the properties of the ice, including enhanced upper ocean mixing in sea ice covered waters (Thomas et al., 2019), sea ice drift (Boutin et al., 2020), and lateral melting of ice floes (Steele, 1992). Hence, the extent to which waves can impact the morphology of the sea ice cover is

defined by the balance between wave energy dissipation as a function of sea ice properties on the one hand, and the break-up of the sea ice by the stresses imposed onto the ice by the waves on the other hand. Evidently, the complex and coupled processes of ice-induced wave attenuation and wave-induced sea ice breakup need to be understood, quantified, and modeled, before wave-ice interaction processes can be reasonably implemented in operational forecasting models.

Studies have, so far, mainly focused on the attenuation of wave energy in sea ice covers and identified a series of conservative

and dissipative processes that damp wave energy in sea ice. These include wave scattering (e.g., Vaughan and Squire, 2007; Meylan and Bennetts, 2018), stresses within the ice layer (e.g., Wang and Shen, 2010; Sutherland et al., 2019), turbulence (Liu and Mollo-Christensen, 1988; Voermans et al., 2019), brine migration (Marchenko and Cole, 2017), and interactions between ice-floes (Rabault et al., 2019; Herman et al., 2019). Although there is still debate regarding when and where these processes are important (Thomson et al., 2018; Squire, 2020), they have been, to various degrees, parameterized, validated,

and/or implemented in numerical wave models (e.g., The WAVEWATCH III Development Group, 2019). Our understanding of wave-induced sea ice breakup is, however, significantly lacking, and few studies are available (with the notable exception of the studies by Crocker and Wadhams, 1989; Langhorne et al., 1998; Dumont et al., 2011; Williams et al., 2013a).

Fundamentally, wave-induced sea ice break-up is determined by a large set of highly environmental dependent wave and ice parameters. Those include the mechanical properties of sea ice (the flexural strength of the ice $\sigma$, elastic or Young's modulus

$Y$), its material properties (ice salinity $S_{ice}$, ice temperature $T_i$, water $\rho_w$ and ice density $\rho_{ice}$), the scale of the ice (ice thickness $h$ and horizontal length scale of the ice $L_{ice}$), as well as wave field characteristics (wave amplitude $a$ and wave length $\lambda$), the gravitational acceleration $g$ and time $t$. We ignore surface tension and viscosity here due to the large length scales associated with the problem, though it is acknowledged that the ice viscosity could potentially play a role. We also ignore $L_{ice}$, the floe size, and focus on solid ice instead, that is, $L_{ice} \gg \lambda$. If we also consider the ice to be flexible enough to follow the wave

surface reasonably well, that is, the ice is not thick enough to be rigid at the length scale of the wavelength, buoyancy effects might be ignored such that $\rho_w$, $\rho_{ice}$, and $g$, are only of minor importance. The ice mechanical properties $\sigma$ and $Y$ are, perhaps, the most complex variables in this set as they are strongly related to the environmental conditions to which the was exposed at its formation and during the rest of its lifetime. In particular, exposure to the cyclic bending of the ice by waves can lower the flexural strength of the ice (e.g., Langhorne et al., 1998), commonly known as fatigue, but can also strengthen the ice when

steady stress loads are applied to the ice (Murdza et al., 2020), such as by wind and currents, whereas local heterogeneities in sea ice can lead to localized concentration of stresses. While these complexities are intrinsic to the physics of the wave-induced sea ice break-up problem, a full understanding of these processes are outside the scope of this study. Here, we ignore the dependence of sea ice material properties with its history (or time $t$), and adopt the traditional dependence of $\sigma$ and $Y$ on the brine volume fraction of the ice $\upsilon_b$, which has been related to the temperature and salinity of the ice, such that $\sigma = f(S_{ice}, T_{ice})$

and $Y = f(S_{ice}, T_{ice})$.





If we then define the wave-induced sea ice break-up similitude by a non-dimensional parameter $I_{br}$ using the Pi-theorem, the break-up problem can be formulated as:

$$I_{br} = f\left(\frac{\sigma}{Y}, \frac{a}{\lambda}, \frac{h}{\lambda}\right). \tag{1}$$

where $\sigma/Y$ is the strain, $a/\lambda$ is the wave steepness and $h/\lambda$ is the relative ice thickness. The dependency of $I_{br}$ on these

parameters can be determined by considering the ice sheet as an elastic plate. This results in the flexural strain

$$\varepsilon = \frac{h}{2}\frac{\partial^2 \eta}{\partial x^2}, \tag{2}$$

where $\eta$ is the wave surface elevation in the horizontal direction $x$. Considering a periodic wave $\eta = a\sin(kx - \omega t)$, where $k = 2\pi/\lambda$ is the wave number and $\omega$ is the radian wave frequency, the maximum strain is defined as (e.g. Dumont et al., 2011):

$$\varepsilon = \frac{2\pi^2 ah}{\lambda^2}. \tag{3}$$

Assuming elastic behaviour of the ice layer, the strain can be considered proportional to the flexural strength $\sigma$ of the ice, leading to $\varepsilon = \sigma/Y$. It then follows that a monochromatic wave will break the ice when $2\pi^2 ahY/\sigma\lambda^2 > 1$. The wave-induced sea ice break-up parameter $I_{br}$ is, therefore:

$$I_{br} = \frac{ahY}{\sigma\lambda^2}. \tag{4}$$

This break-up parameter is consistent with Eqn. (1), and forms the basis of the recent wave-induced sea ice break-up scheme

implemented in coupled wave-ice models (Dumont et al., 2011; Williams et al., 2013a, b; Ardhuin et al., 2018; Boutin et al., 2018, 2020). It follows from Eqn. (3) and Eqn. (4) that the break-up threshold for a monochromatic wave is approximately $I_{br} = 1/2\pi^2 \approx 0.05$, or, strictly speaking, when fatigue and local sea ice heterogeneities are considered $I_{br} \leq 0.05$. Boutin et al. (2018) proposed a threshold 3.6 times smaller, i.e. $I_{br} = 0.014$, based on statistical considerations that the relative maximum strain of a Gaussian random sea state is larger than that of a monochromatic wave. However, to the best of our knowledge,

no study has extensively validated the value of the critical threshold $I_{br}$, nor its universality across a wide range of wave and ice scales. Without convincing validation, the value of this threshold remains an ambiguous extra degree of freedom needed to configure the model and to fit to observations, making it difficult to confidently apply the model at a global scale.

Currently, the lack of a large number of wave-induced sea ice break-up observations, and the uncertainties associated with these, are arguably the foremost reasons for the uncertainty in parameterizing wave-induced sea ice break-up. Measuring wave

and ice properties in the harsh polar environment is challenging, both logistically and technically, even in perfect weather conditions – itself a rare event – especially considering that sea ice break-up often happens during storms. Observing sea ice break-up requires either continuous visual observations, or refined experimental techniques. Even in the event that sea ice



break-up is observed, identification of the exact instant at which the ice breaks (that is, the individual wave responsible for the break-up event) is problematic, as it does not necessarily identify the critical threshold of $I_{br}$, but rather presents a sufficient condition for break-up. That means, that if a wave with known amplitude is observed in the sea ice cover and triggers ice break-up, all what is known is that any wave with the same wave length and an amplitude equal to or larger than the amplitude recorded will break the ice. The contrapositive is true for any wave-induced ice motion taking place without breaking the ice cover. This is further complicated by the deterministic nature of the break-up event itself, that is, in theory we could measure the exact wave event responsible for the break-up, while, in contrast, the identified wave event is a result of the incoherent nature of the wave field and is, therefore, related to the statistical properties of the wave field instead. To bring light on this question, we suggest that many observations of wave-induced sea ice break-up and wave-induced sea ice motion without break-up should be collected. Then, if there should exist a critical, universal threshold for $I_{br}$ as defined in Eqn. (4), a clear separation between unbroken and broken ice conditions should be observed, independently of the details of the ice conditions.

In this study, we attempt to perform such an analysis. For this, we use the results of wave-induced ice motion measurements from two recent field campaigns, one in the Antarctic and the other in the Arctic. In addition, the data obtained are also complemented with an extensive set of observations from both laboratory and field experiments, collected throughout the literature. Thereafter, we approximate the critical wave-induced sea ice break-up criterion based on all data combined, and identify a universal threshold for $I_{br}$.

## 2 Methods

### 2.1 Field Experiments

In the present study, the focus is on data from two recent field experiments, aiming to measure the wave-induced ice motion which lead to sea ice break-up. The first experiment took place in the Antarctic ice pack, and the second in the Arctic ice pack.

#### 2.1.1 Deployment in the Antarctic

The Antarctic deployments occurred on (land)fast ice on the eastern rim of the Amery Ice Shelf (69.3° S 76.3° E, see Fig. 1a) on the 7th of December 2019. The instruments deployed consisted of two wave buoys, denoted as WB in the following (Spotter buoys, from Sofar Ocean Technologies), and two low-cost open-source ice motion loggers (Rabault et al., 2020, hereafter referred to as ice buoys and denoted IB). Both the wave and ice buoys are compact solar-charged position and motion recording instruments with real-time Iridium transmission capability. The wave buoys measure displacement at 2.5 Hz using GPS and transmit wave and position data at a user defined interval. For the deployment period considered here, only integral wave parameters and battery power status were transmitted every half an hour. The ice buoys measure the ice motion using an inertial motion unit (IMU) performing measurements at 10 Hz and transmit the full wave spectrum, geographical location and battery power status at a predefined interval, here, every 3 hours. The accuracy of the vertical displacement is approximately 0.02 m for the wave buoy. For high frequency waves, the accuracy of the ice buoy is $O(mm)$ (Rabault et al., 2016), but the

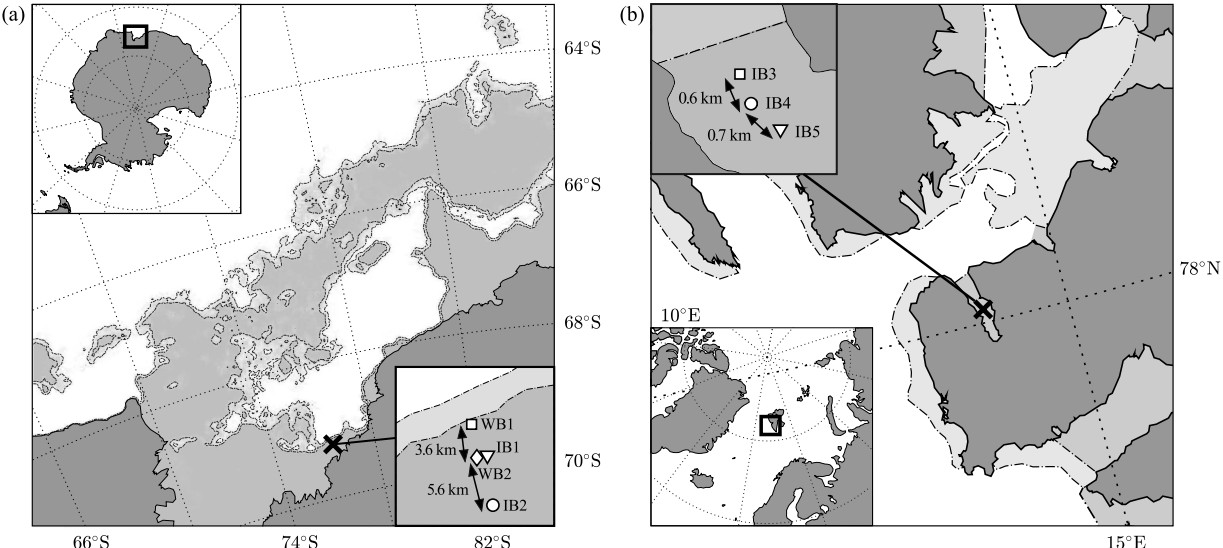

**Figure 1.** Map of the field experiment sites on (a) Antarctic fast sea ice and (b) fast ice in Svalbard. Deployment sites are indicated by a cross. Continents are shaded dark gray, whereas sea ice concentration is represented by the light gray shades using two contour levels, indicative of (a) sea ice concentration of 25% and 75% derived from AMSR2 for 02-Jan-2020 (Spreen et al., 2008), for light and dark grey respectively, and (b) open drift ice and very close drift ice obtained from the Norwegian Meteorological Institute Ice Service for 23-Mar-2020, for light and dark grey respectively. Instruments were deployed along a line perpendicular to the unbroken ice edge (see insets), and consisted of wave buoys (WB: Spotter buoys, Sofar Ocean Technologies) and open source ice motion loggers (referred to as ice buoys, IB; Rabault et al., 2020). Note that in (a), IB1 is shifted laterally for visualization purposes but in reality it is only 40 m apart from WB2.

noise level increases with decreasing wave frequencies (Rabault et al., 2020). For more technical details on the wave and ice
buoys the reader is referred to Raghukumar et al. (2019) and Rabault et al. (2020), respectively.

The instruments were deployed along a line perpendicular to the unbroken ice edge. The first wave buoy (WB1) is about
100–200 m from the edge (see inset Fig. 1a). The second wave buoy (WB2) and first ice buoy (IB1) are deployed 3.7 km from
the solid ice edge, close to each other (the initial distance between WB2 and IB1 is around 40 m), whereas the last ice buoy
(IB2) was deployed about 9.3 km from the edge. Wave buoys were deployed closest to the solid ice edge as these buoys are
capable of surviving in the open water. While the ice buoys have sufficient buoyancy to float, they are expected to malfunction
quickly after entering the water. At the time of deployment, the ice was estimated to be between 1 and 1.2 m thick.

No drift nor significant wave events were recorded for the first three weeks after deployment. On the 2nd of January 2020
the uniform fast ice, on which all instruments rested, broke, and all instruments drifted with the sea ice. In the weeks that
followed, geographical location and vertical ice motion under the influence of waves were obtained until instruments stopped
transmitting. End of transmission happened for IB2 on 22-Jan, for WB1 and WB2 on 1-Feb, and for IB1 on 10-Mar. It is
noteworthy that WB2 reconnected on the 3rd of March for half a day. The wave buoys failed due to depleted batteries, most
likely caused by snow or ice coverage of the solar panels. Considering that batteries of the ice buoys were still close to fully



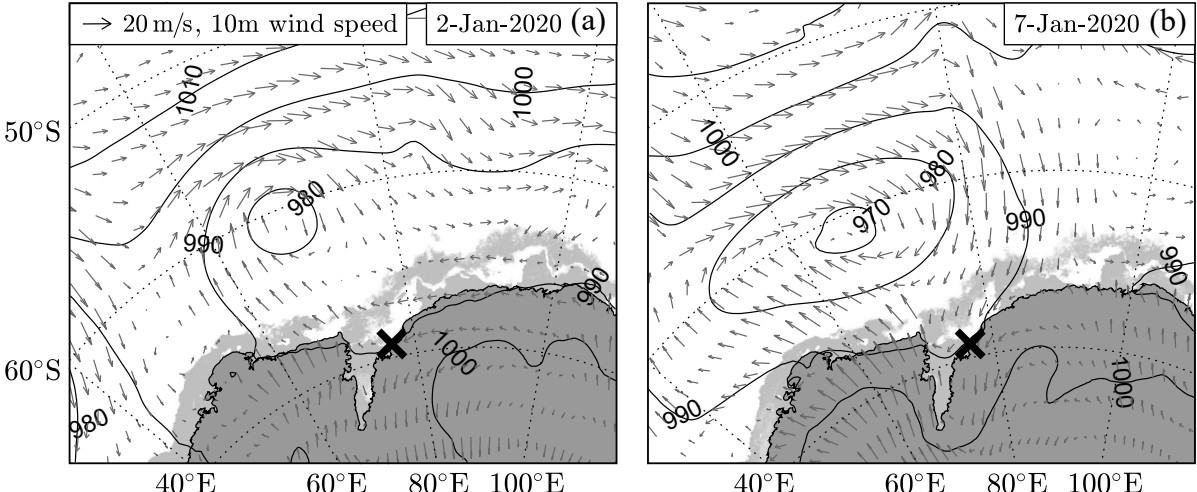

**Figure 2.** Passage of two storms offshore of the Antarctic instrument site on (a) 2-Jan-2020 and (b) 7-Jan-2020. Contours show mean sea level pressure, and wind speed at 10 m is displayed through the vector field (both from ERA5). Light gray shading represents sea ice, derived from AMSR2 data (Spreen et al., 2008). The black cross identifies the deployment site. The presence of relatively high wind speeds over the polynya region on 07-Jan-2020 is expected to generate wind waves at the deployment site.

charged during the last transmissions received from both instruments, we suspect the ice buoys were damaged by the ice or ended being submerged under water between floes. As our interest is in wave-induced sea ice break-up, this study focuses on
observations obtained from January 2–8, which is the period over which initial sea ice break-up was observed for an extensive stretch of fast ice.

During the first week of January, sea ice concentration is well represented by that shown in Fig. 1a. A polynya of approximately 100 km × 300 km separated the fast ice from a 100 km wide band of pack ice. Based on ERA5 re-analysis, three significant low pressure systems passed along the Antarctic continent over the time interval considered. The first merely
skimmed the deployment site on the 2nd of January (Fig. 2a), while the second moved north-east just before reaching the longitude of the instruments around January 5th. The third low-pressure system is expected to have the largest impact on the conditions near the deployment site, with an estimated wind speed of about 10–15 m s$^{-1}$ on the 7th of January (Fig. 2b).

### 2.1.2 Deployment in the Arctic

The second field experiment was performed in Grønfjorden, Svalbard (Fig. 1b). Three ice buoys were deployed on landfast
sea ice between 10 and 13 March 2020, and recovered on the 28th of March. The unbroken ice edge was reasonably stable during the deployment, located roughly halfway through the fjord. The first ice buoy (IB3) was deployed approximately 500 m from the unbroken ice edge. The second (IB4) and third ice buoy (IB5) were deployed 600 m and 700 m apart. Ice thicknesses of 0.3–0.4 m were measured along the main axis of the fjord at the start of the experiment. Based on the water temperature measured just under the ice and on the air temperature, the ice temperature is estimated to be about −8°C. The salinity of the





ice was determined by measuring the conductivity of melted sections of a 0.4 m long ice core, with bulk salinity of 0.68%. Based on visual observations, the ice did not break during this field experiment.

## 2.2 Observations of sea ice break-up in previous literature

In addition to the ice motion and break-up observations collected during our field campaigns, a set of wave-induced break-up data was collected from the literature. Published data were used only when sufficient details about the wave and ice conditions

were presented to determine $I_{br}$. Due to the near absence of concurrent measurements of all wave and ice properties, we consider it to be sufficient when ice thickness, wave height, and wave length (or wave period), are provided. The most critical requirement was that the published sea ice break-up event was, with sufficient confidence, attributed to the observed wave event. We exclude fresh water ice experiments and numerical studies.

The full dataset consists of 31 observations, including 14 wave events that did not result in ice break-up (9 of them from the

laboratory), and 17 events where waves were responsible for the break-up of the ice (7 of which are laboratory observations). Besides the laboratory study of Herman et al. (2018), field observations were taken from Liu and Mollo-Christensen (1988), Cathles et al. (2009), Marchenko et al. (2011), Marchenko et al. (2012), Asplin et al. (2012), Collins et al. (2015), Sutherland and Rabault (2016), Kohout et al. (2016), Marchenko et al. (2019) and Kovalev et al. (2020). We note that the break-up observations made by Liu and Mollo-Christensen (1988) and Kohout et al. (2016) are visual shipborne observations and not *in*

*situ* measurements (see the complete set in Table 1).

In the case of the field experiment of Kovalev et al. (2020), wave conditions resulting in the largest $I_{br}$ were used here as these are the waves most likely responsible for the break-up event observed. For the field observations of Sutherland and Rabault (2016), cracks in the ice were argued to be responsible for the sudden change in the dispersion relation from flexural-gravity waves to gravity waves, and this transition is used here to determine the instant at which the ice was broken by waves.

Additionally, the study of Cathles et al. (2009) is included, and describes the impact of swell on the flexure of the Antarctic ice shelf. Cathles et al. (2009) argue about the potential of most energetic swell events to promote crack propagation of the Nascent Iceberg. In a later study, Massom et al. (2018) showed that there exists a strong correlation between the arrival of swell and the disintegration of the ice shelves. The ice motion amplitudes observed in Cathles et al. (2009) are similar to those measured by Bromirski et al. (2010). While the ice shelf cannot be regarded as a thin ice sheet (and hence the validity of Eqn. (2) for this

event can be questioned), this observation is, nevertheless, included for comparison reasons.

As not all parameters were consistently and/or accurately measured across these studies, the uncertainty of the individual variables were estimated to approximate the uncertainty in $I_{br}$. Each variable was described by a triangular probability distribution, the most likely value of which is typically the value given in the respective study or, alternatively, the mean of the provided range. To obtain an uncertainty for the wave-induced sea ice break-up parameter, a large number of random values

for each variable were generated and the 5th and 95th percentiles of $I_{br}$ were determined.

For the wave amplitude, the most likely value is taken either as the cited wave amplitude, or half the significant wave height measured (i.e. $a = H_s/2$). For the wave period, if no specific period is provided, the (local) peak period is taken. For all direct observations of wave amplitude and wave period an uncertainty of 10% is taken into account as the outer value of the



triangular distribution, while for visual observations we use a larger uncertainty (case specific and dependent on the absolute
values of the variables). Based on the water depth, either estimated or provided, the wave length is calculated following the
linear dispersion relation. The impact of the ice on the wave length (i.e., the flexural, compressive, and ice added mass terms
in the dispersion relation as expressed by for example Sutherland and Rabault (2016)) is assumed to be minor compared to the
uncertainty included in the wave period. This is a reasonable assumption as most measurements have a wave period large than
7 s (e.g., Sutherland and Rabault, 2016; Collins et al., 2018). As measurements of the ice thickness are expected to have higher
uncertainty than the wave properties, an uncertainty of up to 50% is considered, but larger values are chosen for shipborne
visual observations.

The mechanical properties of the ice have the largest uncertainty of all variables involved, in large part, as they are difficult
to measure, particularly in this extreme environment. Only in the studies of Marchenko et al. (2011, 2012, 2019) the flexural
strength ($\sigma$) and/or Young's Modulus ($Y$) were measured *in situ* and therefore provide the narrowest range of uncertainty.
Note that in the case of the tsunami wave observations of Marchenko et al. (2012), details of the ice properties during this
experiment are provided in Marchenko et al. (2013) and Karulina et al. (2019). For the Arctic field experiment (this study) and
the observation of Asplin et al. (2012) only ice salinity and temperature were measured. For these experiments we approximate
$\sigma$ and $Y$ through their strong dependence on brine volume. Using the empirical relation of Frankenstein and Garner (1967),
the brine volume can be approximated by:

$$v_b = S_{ice} \left( \frac{49.185}{|T_{ice}|} + 0.532 \right), \tag{5}$$

where $v_b$ is the brine volume in ppt, $S_{ice}$ is the ice salinity in ppt, and $T_{ice}$ is the ice temperature in °C. This gives an estimated
sea ice brine volume of 4.51% and 6.66% during our Arctic experiment and the study of Asplin et al. (2012), respectively. As
sea ice properties are strongly influenced by the conditions of its formation and development, the empirical relations for sea
ice properties in terms of brine volume are considered to be region-specific (Karulina et al., 2019). Hence, for our Arctic field
experiment we consider empirical relations from the study of Karulina et al. (2019), which is focused on the ice properties in
the Svalbard archipelago, yielding:

$$\sigma = 0.5266 \exp \left( -2.804 \sqrt{v_b} \right), \tag{6}$$

$$Y = 3.1031 \exp \left( -3.385 \sqrt{v_b} \right), \tag{7}$$

where the brine volume is in volume fraction instead of ppt here. The scatter of data for $\sigma$ and $Y$ it is in Karulina et al. (2019)
is used to quantify the uncertainty. For the sea ice break-up observation of Asplin et al. (2012) we use the commonly used
empirical relation of Timco and O'Brien (1994) instead to approximate the flexural strength:

$$\sigma = 1.76 \exp \left( -5.88 \sqrt{v_b} \right). \tag{8}$$



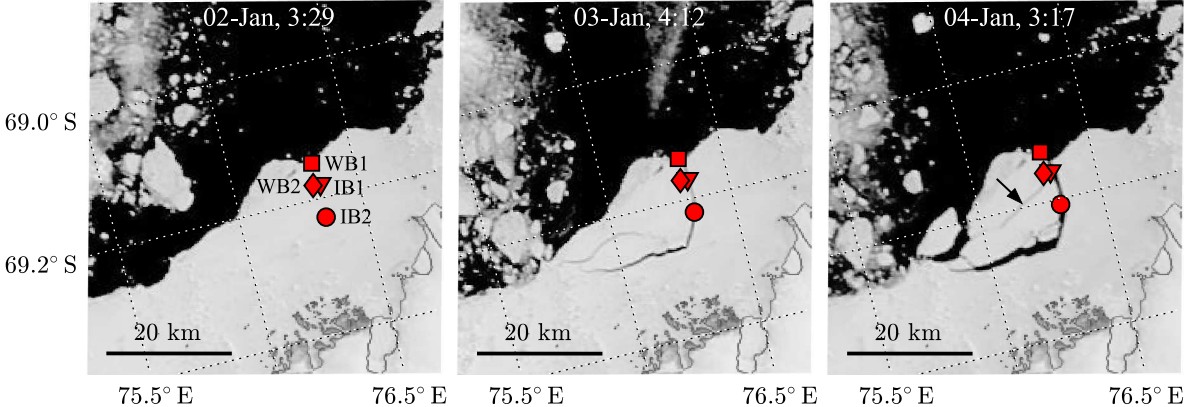

**Figure 3.** MODIS imagery (https://worldview.earthdata.nasa.gov/) of the Antarctic deployment site on three consecutive cloud-free days during the initial sea ice break-up. Instruments are indicated by markers: WB1 (square); WB2 (diamond); IB1 (triangle); IB2 (circle). Note that the marker of IB1 is shifted here for visualization purposes, and that IB1 was originally deployed 40 m from WB2.

For the Young's Modulus we consider the empirical relation of Vaudrey (1977):

$$Y = 5.31 - 0.436\sqrt{v_b}. \tag{9}$$

Note that the unit of brine volume in Eqn. (8) is in volume fraction whereas in Eqn. (9) in ppt. It is worth mentioning that the value for $\sigma$ calculated following this approach in Asplin et al. (2012) is incorrect due to a typographical error in their equation (compare Eqn. (8) here to their Eqn. (4)). An uncertainty of 50% is assigned to $\sigma$ and $Y$ for the observation of Asplin et al. (2012).

For all other observations where no details of sea ice properties were measured or provided, we assign a relatively conservative range of uncertainty to $\sigma$ and $Y$. For experiments within the Svalbard archipelago, we choose a range of $\sigma \in [0.109, 0.415]$ MPa and $Y \in [0.4, 3]$ GPa with most probable values of $\sigma = 2.62$ MPa and $Y = 1.25$ GPa (Karulina et al., 2019). A wider range for $\sigma$ and $Y$ is expected to be found elsewhere and, as such, we expand the uncertainty for observations made in other regions given by $\sigma \in [0.1, 0.7]$ MPa and $Y \in [1, 6]$ GPa with most probable values of $\sigma = 0.4$ MPa and $Y = 3$. A summary of all data used and their estimated uncertainty is provided in Table 1.

# 3 Results

## 3.1 Antarctic deployment

The first break-up event observed during the Antarctic campaign occurred about three weeks after instrument deployment. Based on satellite images, it can be observed that between 02 and 03-01-2020 a giant ice floe (approximately $20 \times 10$ km in size) broke from the fast ice (see Fig. 3). Based on the sudden change in geographical location of all four instruments (not

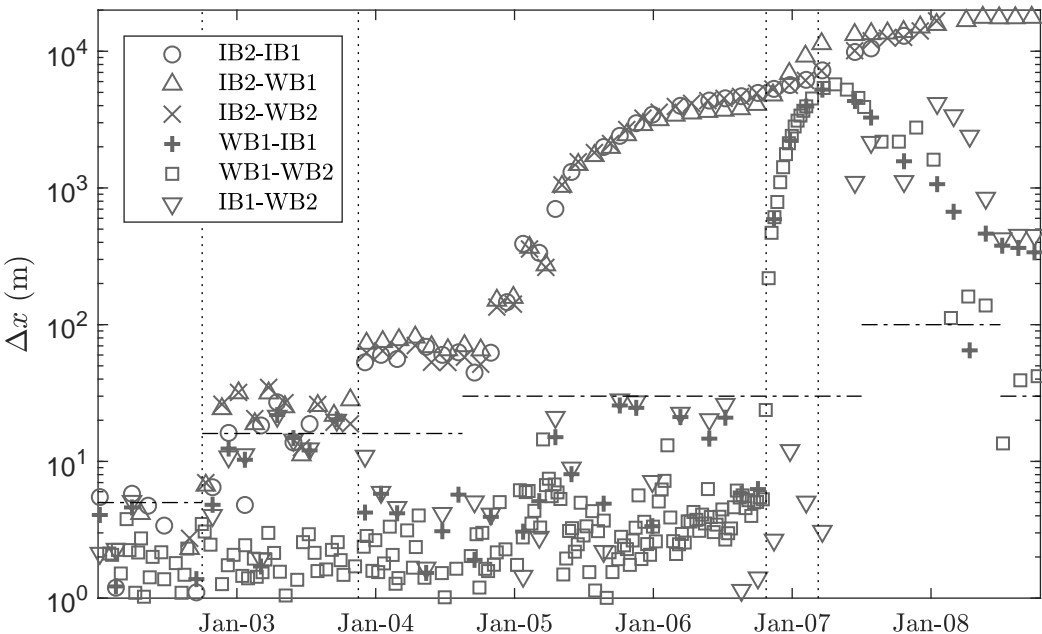

**Figure 4.** Distance $\Delta x$ between instruments relative to their distance at the time of deployment, i.e., $\Delta x$ is taken initially equal to 0, and any change to $\Delta x$ indicates relative motion between the instruments. Vertical dashed lines indicate instances of sea ice break-up: (02-01-2020 18:00) all instruments start drifting due to break-up of large ice mass; (03-01-2020 21:00) IB2 separates from the large ice mass, see also Fig. 3c; (06-01-2020 19:30) WB1 splits from WB2 and IB1; (07-01-2020 04:30) the ice floe holding WB2 and IB1 breaks due to waves generated by the storm depicted in Fig. 2b. Horizontal dashed lines refer to the uncertainty level induced by GPS accuracy and interpolation error, where the latter increases with the drift speed of the instruments.

shown here), this occurred around 01-01-2020 18:00. It also shows that all instruments are located on this giant ice floe which drifted at an average speed of approximately 0.03 m s$^{-1}$ after the initial break-up. Note that on the satellite images of the 03-01-2020 and 04-01-2020 multiple cracks can be observed (see arrow in Fig. 3). Unfortunately, clouds in the days after prevent us from monitoring the ice conditions in the days that followed.

     As all instruments transmit their geographical location at regular intervals, albeit at different times, we can identify the

occurrence of sea ice break-up and approximate the times at which these events occurred through the monitoring of sudden changes in the relative distance between buoy-pairs $\Delta x$ during the deployment (see Fig. 4). In all the following, the distance $\Delta x$ is relative to the distance at the time of deployment, i.e., initially $\Delta x$ is taken equal to 0, and any change in $\Delta x$ is due to relative motion of the instruments. However, for brevity, we will refer to this quantity as the 'distance' between the instruments.

     As the geographical coordinates of the instruments are not transmitted at the same time and interval, we linearly interpolate

the latitude and longitude coordinates to match between buoy-pairs. As the ice floe upon which the instruments rest drifts,





interpolation of the geographical location introduces a maximum error of typical magnitude $|\delta| \approx (\Delta t^2/8) \max|\Delta x''(t)|$, where $\Delta t$ is the data transmission interval. The estimated value of the error $\delta$ is indicated by the horizontal dashed lines in Fig. 4. Before the first sea ice break-up event, the approximate maximum error of $\Delta x$ is 5 m, a result of the accuracy of the GPS units when kept stationary during the initial three weeks of the deployment. From the instant at which the giant ice floe breaks

from the ice cover and starts drifting (02-01-2020 18:00), the error increases to typically $\delta = 16$ m. Note that the distance between all buoy-pairs remains constant just after the separation event of the giant ice floe, as all instruments remain on the one ice floe. Also note that the accuracy of the distance between the two wave buoys is considerably better than with other buoy-pairs, as the data transmission interval $\Delta t$ is considerably smaller for the wave buoys than for the ice buoys.

After the giant ice floe separates from the ice cover and starts drifting, the next break-up event is thought to occur around

the 03-01-2020 at 21:00, where the distance between IB2 and the other three buoys instantly increases to a distance of 60–70 m (Fig. 4). This is in line with the satellite imagery (Fig. 3), where on 03-01-2020 the crack does not seem to have propagated all the way eastward, whereas on the 04-01-2020 the crack seems to have split the giant ice floe completely (see the arrow, Fig. 3). It is not until the 05-01-2020 that the distance between IB2 and the other instruments increases further. The third break-up event occurred around 06-01-2020 19:30, where the northernmost deployed instrument, WB1, splits from WB2 and IB1 (Fig.

3). This is followed shortly after by a fourth break-up event occurring around 07-01-2020 4:30 where the distance between WB2 and IB1 increases to about a kilometer within just 3 hours.

To determine whether these break-up events were caused by wave-induced flexural motion, they are compared against the wave motions recorded by the instruments. Fig. 5 shows the significant wave height and peak wave period measured by the instruments over a duration of six days after the initial break away of the giant ice floe. Note that up to 05-01-2020, the

instruments do no provide reliable wave information as recorded motions are below the noise threshold of the instruments. While this can be observed indirectly from the transmitted $H_s$ and $T_p$, for the ice buoys this is confirmed through observation of the wave energy spectra, showing a linear energy decay in log-scale from low to high frequencies, which corresponds to the noise threshold of the IMU (Rabault et al., 2020). There are, however, two clear instances of coherent measurements of both the peak period and wave height, see shaded areas in Fig. 5.

For the first break-up event on the 02-01-2020, no waves were measured above the noise level of the instruments and the cause of the break away of the giant ice floe remains speculative. ERA5 re-analysis data just north of the most northern sea ice edge indicates the presence of a 3 m swell a few hours preceding the break-up (generated by the storm depicted in Fig. 2a), and, as such, swell might have been a potential cause of the break-up. However, as no significant ice motion events were recorded during this period of time by the instruments, it suggests that this swell event was largely dissipated by the vast sea

ice band in front of the polynya. As there are no reliable wave measurements for the second and third break-up events either, we can only speculate about the cause of these events as well. As a few large cracks in the giant ice floe are already visible on the 03-01-2020 (Fig. 3), therefore, it is most likely that the second break-up event was initiated at the same instant at which the giant ice floe broke from the fast ice cover. The third break-up event, however, is most likely induced by waves generated by the more energetic storm passing the deployment site during this part of the deployment (Fig. 2b).

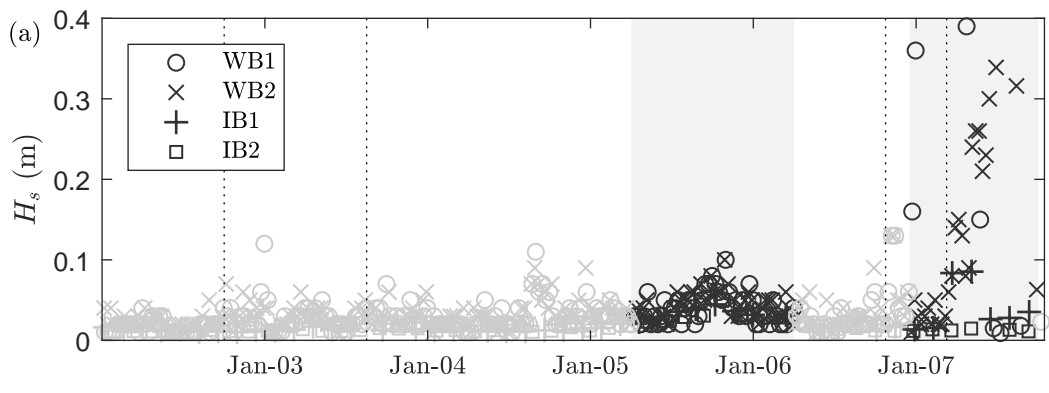

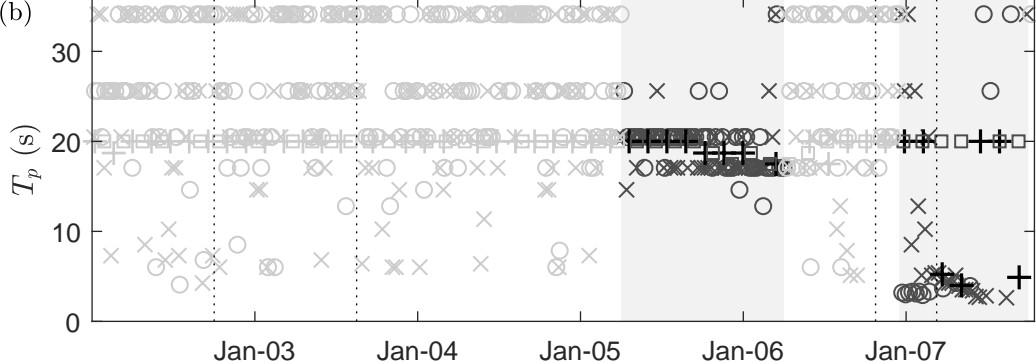

**Figure 5.** (a) Significant wave height and (b) peak period measured by the four instruments during the break-up of the Antarctic fast sea ice cover. Based on consistency of the measured peak period between all instruments, two sections contain reliable wave measurements over noise thresholds, corresponding to a swell event with maximum ice motion obtained on 05-01-2020 18:00, and wind waves just after 07-01-2020. These sections correspond to the greyed areas and dark markers. Note that the vertical dashed lines indicate sea ice break-up events, extracted from Fig. 4.





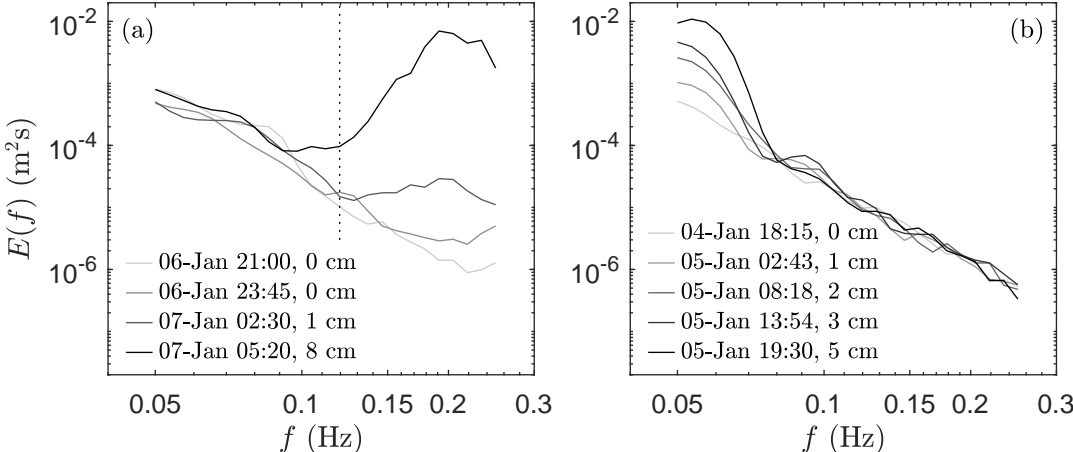

**Figure 6.** Wave energy spectra measured by (a) IB1 during the fourth identified sea ice break-up event, and (b) IB2 during a swell event without sea ice break-up. The significant wave height of the spectra is provided in the legend, note, for (a) only the high frequency part (i.e. $f > 0.13$) of the spectrum is considered, see dashed line. The spectra obtained on the 06-01-2020 21:00 (a) and 04-01-2020 18:15 (b) correspond to the noise level of the IMU. Both the measured wind waves in (a) and swell in (b) are well above noise levels. Note that in (a) energy in the high frequency part of the spectrum increases substantially after ice breakup, which is estimated to take place around 07-Jan 4:30.

Unlike the first three break-up events, wave motions above noise thresholds were measured during the fourth sea ice break-up event. In particular, this break-up event coincides with the passage of the low pressure system and the presence of high wind speeds of about 10–15 m s$^{-1}$, over and aligned with the main axis of the polynya region (this based on ERA5, see Fig. 2b). With an area of approximately $100 \times 300$ km, the polynya provide sufficient fetch for the waves to develop. Around the time of break-up, a consistent peak wave period of around 5 s is measured by WB2 and, to lesser extent, by IB1. The wave energy

spectra measured by WB2, however, shows that the wave energy in the high frequency range (around $f = 0.2$ Hz) increases steadily with time (Fig. 6a). This explains for the sudden change in $T_p$ for IB1: the noise level at the lowest resolved frequency is larger than the measured wave energy in the high frequency range, so the wave amplitude of the relatively high-frequency waves has to reach a threshold before it is considered as the peak wave frequency $T_p$.

The significant wave height of the high frequency waves (that is, when considering the wave energy for $f > 0.12$ Hz) is only

0.01 m at 2:30 on 07-01-2020, and 0.08 m at 5:20. This suggests that the fourth break-up event, occurring around 4:30, was induced by waves with period of approximately 5 s, with an estimated wave height of around 0.04 m. It is noteworthy that the wave buoy WB1, which separated from WB2 and IB1 during the third break-up event, measured a significant wave height of up to 0.4 m at the time of the fourth break-up event, also with a period of approximately 4–5 s, indicating that the energetic wind waves were generated locally (since, if generated in the Southern Ocean, these waves would have dissipated rapidly in

the sea ice band north of the polynya).





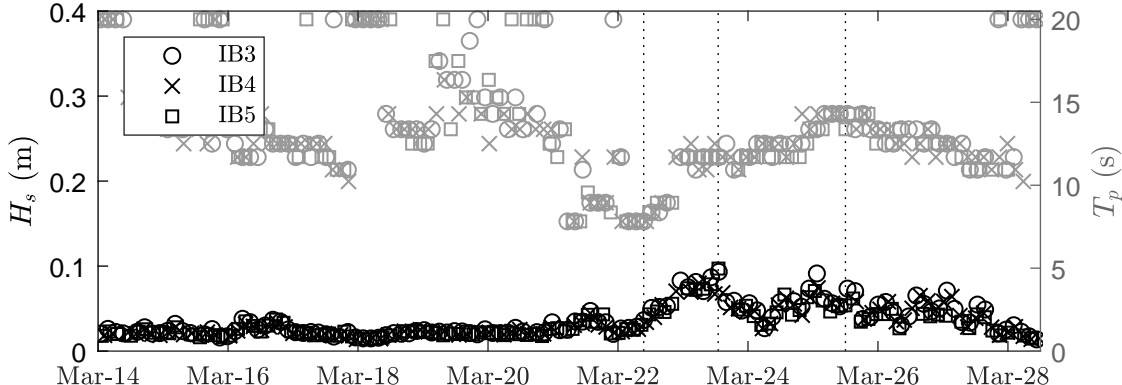

**Figure 7.** Significant wave height and peak period measured by three ice buoys deployed on fast ice in Svalbard. The dashed lines identify three events with distinct peak wave period and peak significant wave height.

Besides this wave-induced break-up event, a distinct swell event around the 05-01-2020 18:00 was measured by all four instruments (Fig. 5), though, it did not lead to sea ice break-up. From the spectra measured by the ice buoys it can be seen that the observed wave energy is comfortably above instrument noise level (Fig. 6b). The time frame of this swell event corresponds well to the passage of a storm moving north-east at this instant. This swell event will be used as a non-break-up event with a
significant wave height of 0.05 m and period $T = 17 - 20$ s (Fig. 5 and 6b).

## 3.2 Arctic experiment

During the Arctic field campaign, no sea ice break-up was observed and all instruments remained stationary during the deployment. The measurements of significant wave height and peak wave period are shown in Fig. 7. Three distinct wave events are considered as ice motion observations without sea ice break-up. The events have a peak period $T_p = 7.8$, 11.7 and 14.3 s
respectively, and corresponding wave heights are $H_s = 4$, 0.10 and 0.07 m (see dashed lines in Fig. 7).

## 3.3 Ice break-up threshold

Combining the break-up and non-break-up events obtained during the two field campaigns, and the set of existing published observations, the ice break-up parameter $I_{br}$ can be determined (see results in Fig. 8). In Fig. 8, we plotted $I_{br}$ against the relative ice thickness $h/\lambda$ to separate between ice breaking and non-breaking observations. Note that the red markers identify
events where the ice remained intact under the wave motion. We reiterate that, similarly but contrapositive for the unbroken ice events, observations of sea ice break-up define a sufficient condition for wave-induced sea ice break-up, not the absolute





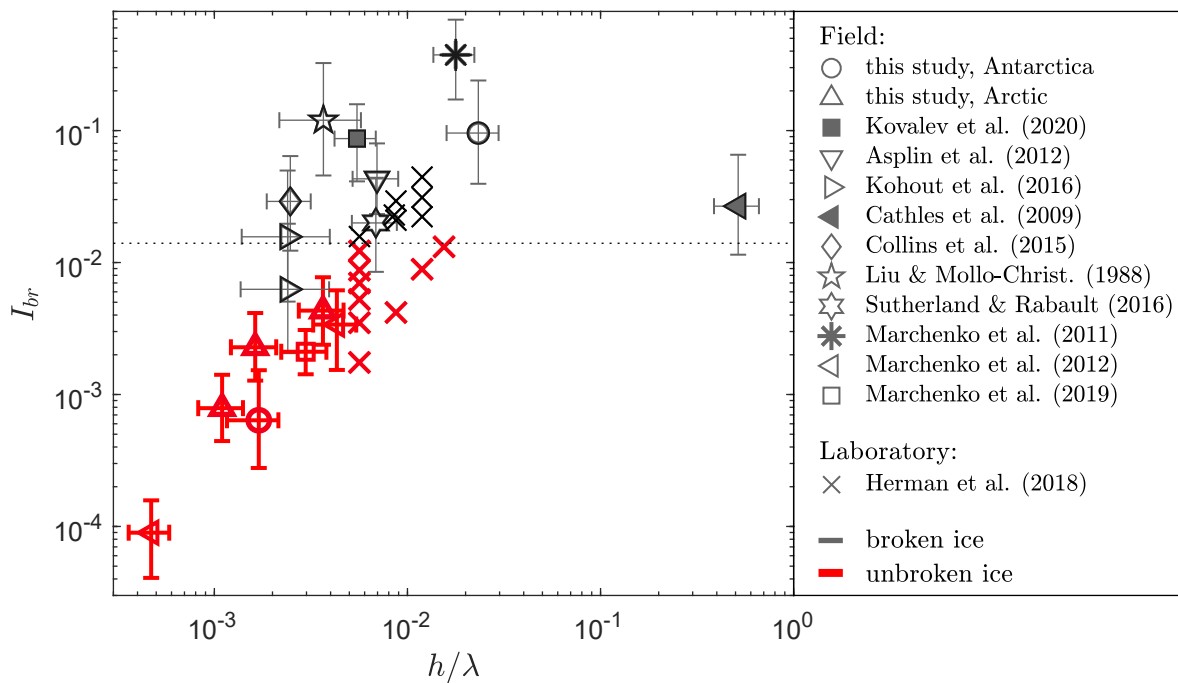

**Figure 8.** Observations of $I_{br}$ against the relative ice thickness $h/\lambda$ for the complete data set. Events of wave-induced sea ice break-up are indicated with black markers, whereas observations where the flexural motion did not lead to break-up of the sea ice are shown with red markers. The observational threshold value $I_{br} \approx 0.014$, that separates the break-up from the non-break-up events, is indicated by the dashed line.

threshold for the break-up parameter $I_{br}$. It is seen that broken and unbroken observations can be reasonably separated by a constant value of $I_{br}$. Therefore, based on the data presented in Fig. 8, we find the critical value of $I_{br}$ to be equal to:

$$I_{br} \approx 0.014. \tag{10}$$

While this threshold is most accurately boxed by the laboratory experiments of Herman et al. (2018) (particularly as these constitute about half of the points in the data set), observations obtained in the field are well aligned with this threshold too. Note that, while one of the shipborne break-up observations of Kohout et al. (2016) falls below this threshold, the large uncertainty of this particular visual observation covers both sides of the critical threshold.

## 4   Discussion

In the present work, we have collected experimental observations, both from the laboratory and the from field, displaying both wave-induced sea ice break-up events and wave-induced ice motion events without break-up. Thereafter, we have used these data to estimate the critical threshold value for the wave-induced sea ice break-up parameter $I_{br}$. We find that observations



consistently point to a constant value of $I_{br}$, which we estimate to be $I_{br} \approx 0.014$ (see Fig. 8). Note, however, as we can only measure statistical wave properties in an incoherent wave field, and thus $I_{br}$ is a probabilistic metric rather than a deterministic,

the threshold observed in this study therefore suggests that above $I_{br} = 0.014$ the ice is very likely (but not necessarily) going to break. Though the data set is still rather limited, it is promising that both field and laboratory observations are well aligned with this critical value. In particular, laboratory-grown ice is known to have distinctly different material properties (e.g., Herman et al., 2018; Squire, 2020) where, for instance, the ice in the laboratory data used here has a critical strain one to two orders of magnitude larger than that of sea ice in the field. Interestingly, the swell-induced crack propagation of the Ross Ice Shelf, as

implied by Cathles et al. (2009) and Massom et al. (2018), seems to fit well within the overall dataset, indicating that it might be possible to extrapolate the wave-induced sea ice break-up criterion to much thicker ice covers as well.

While we observe that the critical value determined in this study is three to four times smaller than that of a monochromatic wave (e.g. Williams et al., 2013a), this value is remarkably similar to that proposed by Boutin et al. (2018), who argued, based on statistical considerations, that a factor of 3.6 should be used to take into account the random nature of the wave field and the

resulting stochastic distribution in individual wave amplitudes. However, as the ice in the laboratory experiments of Herman et al. (2018) were exposed to monochromatic waves, rather than a random wave field, it remains uncertain whether this factor is indeed a statistical correction, a compensation for the simplification of the sea ice material properties (that is, by ignoring fatigue and the presence of sea ice heterogeneities, the critical flexural strength of the ice is effectively lower than those values used here), or, more likely, a combination of both. Either way, our experimental results are in support of the current approaches

developed to model the break-up of a solid ice cover under wave forcing in coupled numerical models, albeit further study is required to understand the finer details of the physics behind wave-induced sea ice break-up.

Although the wave-induced sea ice break-up parameter $I_{br}$ seems to be physically sound, the scaling of $I_{br} \propto h$ is problematic when the ice material properties $Y$ and $\sigma$ remain virtually unchanged when thin ice is considered, that is, an infinitely thin ice sheet becomes numerically unbreakable (as noted in: The WAVEWATCH III Development Group, 2019). However,

for small ice thicknesses, other physical processes may be naturally dominant, such as compressive or tensile failure of the ice through wind and ocean current shear forces. Indeed, the relative effect of such forcing scales inversely to the ice thickness (e.g., Mellor, 1986), contrary to what is obtained with the present expression for $I_{br}$. This highlights that waves and sea ice are part of a complex coupled system at the interface of the atmosphere and ocean, and that many different physical phenomena influence sea ice dynamics. Waves can, however, still play a critical role in the break-up of thin ice. For instance, thick ice

attenuates wave energy more strongly than thin ice (e.g., Doble et al., 2015; Meylan et al., 2018; Liu et al., 2020), therefore, thin ice is generally exposed to more wave energy, including shorter wavelengths. Moreover, there are still significant uncertainties in the actual mechanical properties of very thin ice relative to thicker ice. Fast grown thin ice (for instance, in the case of very cold air temperature) has a lower flexural strength compared to slow grown ice (Bond and Langhorne, 1997), which the literature claims to be caused by its higher bulk salinity (Perovich and Richter-Menge, 1994). Moreover, Kovacs (1996) finds

that the salinity of young ice decreases with increasing ice thickness, implying that thin ice might be consistently weaker than thicker ice following Eqn. (6) and (8). As ice properties can vary significantly in time, more studies are required to accurately measure and define the mechanical properties of sea ice in terms of more readily available air-sea-ice properties, and the role



of ice inhomogeneities caused by bubbles and brine pockets, ice ridges, pools, and ice thickness variability, needs to be further investigated.

Field observations of waves, ice motion, ice material properties, and sea ice break-up identification, bring unavoidable uncertainties, resulting in a significant uncertainty for $I_{br}$. Particularly the mechanical properties of sea ice are uncertain due to the validity of the experimental methods used (e.g., see Timco and Weeks, 2010; Karulin et al., 2019), fatigue (e.g., Langhorne et al., 1998), spatial heterogeneity at various scales and even questions regarding the scaling effects of the ice flexural strength (Aly et al., 2019). Identifying the instant at which the ice breaks creates an additional uncertainty. The method which consists

in identifying the instant of sea ice break-up through the spatial divergence of instrumentation, as applied in this study, is not foolproof by itself. In fact, if the ice floes do not drift apart after break-up, the relative distance between instruments will not change. As the sea ice in our field experiments was drifting during break-up, it is expected that the resulting ice floes after break-up will attain a different drift speed. In the case of the Antarctic field campaign, the instruments drifted at a speed of 0.03–0.20 m s$^{-1}$ and, even if the differential drift between floes immediately after break-up is only a fraction of this drift speed,

this will be noticed from the position of the instruments within hours of the time of break-up, at most.

    Therefore, a dedicated field experiment, with the aim to closely monitor both the mechanical properties of the ice and the exact instant at which the ice breaks, is highly desirable and is expected to provide further clarification over the accuracy of the observed threshold for $I_{br}$ reported here. Until then, many more observations of wave-induced ice motion leading up to ice break-up are necessary to further substantiate the wave-induced break-up parameter and its critical threshold. Evidently,

development of low-cost and open source instrumentation is critical in obtaining a large dataset of break-up observations, as it promotes the deployment of ice buoys in larger quantities and, therefore, allows to dramatically increase the overall volume of data reporting the interactions between sea ice, waves, atmosphere, and the ocean.

## 5   Conclusions

We presented observations of wave-induced ice motion and sea ice break-up events from two field experiments, one in the

Antarctic and the other in the Arctic. Using the relative displacement between the instruments deployed, four sea ice break-up events were registered in the Antarctic field experiment, although only one could, with reasonably certainty, be linked to waves. While no sea ice break-up events were observed in the Arctic field experiments, it provided three wave events without sea ice break-up. We used these observations, supplemented with existing data taken from a wide body of the literature, to reach an estimate for the critical threshold of the wave-induced sea ice break-up parameter $I_{br} = ahY/\sigma\lambda^2$, where $a$ is the

wave amplitude, $h$ is the ice thickness, $Y$ is the Young's Modulus, $\sigma$ is the ice flexural strength, and $\lambda$ is the wave length. We find that a value $I_{br} = 0.014$ separates well observations of wave-induced break-up and non break-up events. Observations include laboratory measurements, as well as suspected cracking of the Antarctic ice shelf. The physical relevance of $I_{br}$ is substantiated by the diversity of cases present in the data, from laboratory to the field, the Antarctic to the Arctic, and thin ice to very thick ice. However, significantly more observations of sea ice break-up are necessary and, perhaps, more sophisticated



measurement techniques need to be developed, in order to identify the exact instant at which break up occurs, and the wave conditions responsible for the observed sea ice break-up.

*Data availability.*  Data will be made available in a public repository.

*Author contributions.*  JV and AB conceptualized. JV and JR built instrumentation. KF and IR deployed and retrieved instrumentation. MD and GS assisted in satellite imagery. AB and AM administered the project. JV and JR prepared original draft. All authors reviewed and edited

the manuscript.

*Competing interests.*  The authors declare that there is no conflict of interest.

*Acknowledgements.*  We acknowledge the use of imagery from the NASA Worldview application (https://worldview.earthdata.nasa.gov/), part of the NASA Earth Observing System Data and Information System (EOSDIS). Authors would like to thank the crew of AARI for their assistance during the deployment of the instruments used in the Antarctic. We thank Prof. Atle Jensen and Ing. Olav Gundersen for their

support in assembling IB1 and IB2 (IB3, IB4, and IB5 were assembled at Melbourne University following the same design). Data collection in Grønfjorden, Svalbard was conducted within the expedition 'Spitsbergen-2020' organised by Russian Scientific Arctic Expedition on Spitsbergen Archipelago (RAE-S), AARI. JJV and AVB acknowledge support from the Joyce Lambert Antarctic Research Fund (Grant 604086); JJV, AVB, and PH were supported by the Australian Antarctic Program under Project 4593 plus PH under 4506; JJV, JR, KF, AM, and AVB acknowledge the support of the Research Council of Norway through the SFI SIB project. JR was supported in the context of DOFI

project (Univ. of Oslo, Grant Number 280625).



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





**Table 1.** Wave and ice properties used to calculate the break-up number $I_{br}$ for observed wave-ice events. Uncertainty in variables is taken into account through a triangular probability distribution, defined by the minimum, most probable and maximum value, respectively. Notes: [1] while break-up was observed by the authors, Herman et al. (2018) argue this was induced by reflecting waves rather than the incoming waves and, as such, this experiment is treated as a non-break-up event; [2] water depth is not required to determine the characteristic wave length as this is given by the authors directly; [3] Due to a typographical error in Asplin et al. (2012) their estimated flexural strength $\sigma = 40.7$ KPa is incorrect, the correct value should be 0.39 MPa.

|  | ice status | $H_s$ (m) | $T$ (s) | $\lambda$ (m) | $d$ (m) |
|---|---|---|---|---|---|
| this study, Antarctica | break-up | $0.04 \pm 0.01$ | $5 \pm 0.5$ | $(32, 39, 47)$ | 450 |
| this study, Antarctica | no break-up | $0.05 \pm 0.005$ | $18.5 \pm 1.5$ | $(451, 534, 624)$ | 450 |
| this study, Arctic | no break-up | $0.1 \pm 0.01$ | $11.7 \pm 1.2$ | $(173, 214, 258)$ | 138 |
| this study, Arctic | no break-up | $0.04 \pm 0.004$ | $7.8 \pm 0.8$ | $(77, 95, 115)$ | 138 |
| this study, Arctic | no break-up | $0.07 \pm 0.007$ | $14.3 \pm 1.4$ | $(258, 317, 378)$ | 138 |
| Kovalev et al. (2020) | break-up | $0.16 \pm 0.016$ | $8 \pm 0.8$ | $82 \pm 11$ | 15.3 |
| Asplin et al. (2012) | break-up | $0.8 \pm 0.08$ | $13.5 \pm 1.4$ | $(230, 285, 344)$ | 1000 |
| Kohout et al. (2016) | break-up | $(0.25, 0.5, 1)$ | $15 \pm 3$ | $(225, 351, 506)$ | 1000 |
| Kohout et al. (2016) | break-up | $(0.1, 0.2, 0.4)$ | $15 \pm 3$ | $(225, 351, 506)$ | 1000 |
| Cathles et al. (2009) | break-up | $0.006 \pm 0.0006$ | $17.5 \pm 1.8$ | $(387, 478, 579)$ | 600 |
| Collins et al. (2015) | break-up | $0.9 \pm 0.09$ | $12 \pm 1.2$ | $(181, 220, 261)$ | 80 |
| Liu and Mollo-Christensen (1988) | break-up | $2 \pm 0.5$ | $18 \pm 1.8$ | $(188, 250, 313)$ | [2] |
| Sutherland and Rabault (2016) | break-up | $0.08 \pm 0.008$ | $7.14 \pm 0.7$ | $80 \pm 16$ | 85 |
| Marchenko et al. (2011) | break-up | $0.2 \pm 0.02$ | $7 \pm 0.7$ | $28 \pm 3$ | 1.75 |
| Marchenko et al. (2012) | break-up | $0.31 \pm 0.03$ | $93 \pm 9.3$ | $1990 \pm 200$ | 47 |
| Marchenko et al. (2012) | break-up | $0.14 \pm 0.014$ | $12.6 \pm 1.3$ | $217 \pm 32$ | 47 |
| Marchenko et al. (2019) | no break-up | $0.03 \pm 0.003$ | $8 \pm 0.8$ | $(81, 100, 121)$ | 160 |
| Herman et al. (2018) | no break-up | 0.02 | 1.27 | 2.52 | [2] |
| Herman et al. (2018) | break-up | 0.05 | 1.27 | 2.52 | [2] |
| Herman et al. (2018) | break-up | 0.07 | 1.27 | 2.52 | [2] |
| Herman et al. (2018) | break-up | 0.1 | 1.27 | 2.52 | [2] |
| Herman et al. (2018) | break-up | 0.1 | 1.5 | 3.51 | [2] |
| Herman et al. (2018) | no break-up | 0.01 | 2 | 6.17 | [2] |
| Herman et al. (2018) | no break-up | 0.01 | 1.6 | 3.99 | [2] |
| Herman et al. (2018) | no break-up | 0.01 | 1.2 | 2.25 | [2] |
| Herman et al. (2018) | no break-up | 0.02 | 2 | 6.17 | [2] |
| Herman et al. (2018) | no break-up | 0.03 | 2 | 6.17 | [2] |
| Herman et al. (2018) | no break-up | 0.04 | 2 | 6.17 | [2] |
| Herman et al. (2018) | no break-up | 0.05 | 2 | 6.17 | [2] |
| Herman et al. (2018) | no break-up[1] | 0.07 | 2 | 6.17 | [2] |
| Herman et al. (2018) | break-up | 0.09 | 2 | 6.17 | [2] |
| Herman et al. (2018) | break-up | 0.05 | 1.6 | 3.99 | [2] |
| Herman et al. (2018) | break-up | 0.07 | 1.6 | 3.99 | [2] |



| $h$ (m) | $Y$ (GPa) | $\sigma$ (MPa) |
|---|---|---|
| $(0.5, 1, 1.2)$ | $(1, 3, 6)$ | $0.4 \pm 0.3$ |
| $(0.5, 1, 1.2)$ | $(1, 3, 6)$ | $0.4 \pm 0.3$ |
| $0.35 \pm 0.12$ | $(1, 1.5, 3)$ | $0.29 \pm 0.11$ |
| $0.35 \pm 0.12$ | $(1, 1.5, 3)$ | $0.29 \pm 0.11$ |
| $0.35 \pm 0.12$ | $(1, 1.5, 3)$ | $0.29 \pm 0.11$ |
| $0.45 \pm 0.15$ | $(1, 3, 6)$ | $0.2 \pm 0.07$ |
| $2 \pm 0.67$ | $(0.88, 1.75, 2.63)$ | $0.39 \pm 0.2^{(3)}$ |
| $(0.375, 0.75, 1.5)$ | $(1, 3, 6)$ | $0.4 \pm 0.3$ |
| $(0.375, 0.75, 1.5)$ | $(1, 3, 6)$ | $0.4 \pm 0.3$ |
| $250 \pm 85$ | $(1, 3, 6)$ | $0.4 \pm 0.3$ |
| $0.55 \pm 0.18$ | $(0.4, 1.25, 3)$ | $0.26 \pm 0.15$ |
| $(0.4, 0.8, 1.6)$ | $(1, 3, 6)$ | $0.4 \pm 0.3$ |
| $0.55 \pm 0.18$ | $(0.4, 1.25, 3)$ | $0.26 \pm 0.15$ |
| $0.5 \pm 0.17$ | $(0.4, 1.25, 3)$ | $0.25 \pm 0.05$ |
| $0.94 \pm 0.31$ | $(0.2, 0.77, 1.6)$ | $0.339 \pm 0.034$ |
| $0.94 \pm 0.31$ | $(0.2, 0.77, 1.6)$ | $0.339 \pm 0.034$ |
| $0.3 \pm 0.1$ | $(1.32, 1.61, 1.89)$ | $0.335 \pm 0.025$ |
| $0.03$ | $0.009$ | $0.048$ |
| $0.03$ | $0.009$ | $0.048$ |
| $0.03$ | $0.009$ | $0.048$ |
| $0.03$ | $0.009$ | $0.048$ |
| $0.03$ | $0.009$ | $0.048$ |
| $0.035$ | $0.016$ | $0.042$ |
| $0.035$ | $0.016$ | $0.042$ |
| $0.035$ | $0.016$ | $0.042$ |
| $0.035$ | $0.016$ | $0.042$ |
| $0.035$ | $0.016$ | $0.042$ |
| $0.035$ | $0.016$ | $0.042$ |
| $0.035$ | $0.016$ | $0.042$ |
| $0.035$ | $0.016$ | $0.042$ |
| $0.035$ | $0.016$ | $0.042$ |
| $0.035$ | $0.016$ | $0.042$ |
| $0.035$ | $0.016$ | $0.042$ |