# Peer review of "Experimental evidence for a universal threshold characterizing wave-induced sea ice break-up"

_The Cryosphere, 2020_

## Referee Comment (RC1) · Anonymous Referee #1 · 1 Sep 2020

This paper presents an analysis of recent field campaigns in which the wave conditions and simultaneous sea ice break up were measured. This data is compared with data from previous observations to determine a threshold for breaking. Such a threshold will be useful provided it is accurate. Even if it is not entirely accurate, I believe it will serve as a valuable benchmark for comparison. I am supportive of the publication.

The fundamental difficulty in this from a theoretical point of view is that the sea state is random with a range of periods. It is therefore difficult to assign to any break-up event a single value for $\lambda$ unless it is for a wave tank experiment. This point should be discussed.

[Figure]

The equations for Young's modulus etc. are essential and summarise literature which is not well known. Multiple authors, mostly those associate with Squire, have used 6GPa for the Youngs modulus, which is an overestimate. However, it should be explained clearly what the units in the formulae are, and the units should be made consistent is possible (e.g. the units of brine volume).

The breaking model, in fact, contains two contradictions/paradoxes. One is that in the limit of small thickness ice is unbreakable, and the other is that short-wavelength waves will break any ice. The second point seems to have been missed by the authors. However, the model assumes that the ice is moving compliantly with the sea surface and the wavelengths are so long that the sea ice can be modelled as a negligible surface. Some discussion of this point and the regime in which it is valid would be useful.

The literature review is mostly complete. However, the first coupled attenuation and breaking model appeared in Kohout AL, Meylan MH. An elastic plate model for wave attenuation and ice floe breaking in the marginal ice zone. *Journal of Geophysical Research: Oceans*. 2008 Sep;113(C9). The authors concluded that their attenuation model was failing because of the overprediction of the break-up.

---

## Referee Comment (RC2) · Anonymous Referee #2 · 24 Sep 2020

**REVIEW OF 'EXPERIMENTAL EVIDENCE FOR A UNIVERSAL THRESHOLD CHARACTERIZING WAVE-INDUCED SEA ICE BREAK-UP'**

**VOERMANS ET AL**

**1. General comments**

This is a very nice collection of observations and the main result of figure 8 is surprising, but convincing with quite a few events included that represent many different scales and seem to have been careful with uncertainties. I recommend publication with minor revisions. I have a small suggestion about the definition of the $I_{br}$ threshold to make it more intuitive.

**2. Specific comments**

(1) p3: section about $I_{br}$ could be improved - I have never heard the term "similitude" or of the "Pi-theorem" before - can you think of a better name? Using $ka$ (steepness) and $kh$ instead of converting $k$ to $\lambda$ would get rid of many factors of $2\pi$, and $I_{br}$ could become a stress relative to the flexural strength or a strain relative to the breaking strain for a beam. (The critical value would be about $4\pi^2 \times 0.014 = .55$ I guess). Since the relationship looks like it could have some universality it is worth presenting it somewhat more intuitively.

(2) "sheet as an elastice plate" — "sheet as a thin elastic plate" (or maybe simply an elastic beam, since you are using the $\sigma = Y\varepsilon$ relation below).

(3) p16: "infinitely thin ice sheet becomes numerically unbreakable" — the opposite problem is that the strain as $kh \to \infty$ (shorter waves/thicker ice) also becomes infinite. In that case including reflection by ice edges is one way to reduce the strain inside the ice [1, 2]. Using the ice wavelength instead of the open water one could also make a difference here too. For both points the ice sheet example of Cathles et al jumps to mind.

**References**

[1] Guillaume Boutin, Fabrice Ardhuin, Dany Dumont, Caroline Sévigny, Fanny Girard-Ardhuin, and Mickael Accensi. Floe size effect on wave-ice interactions: Possible effects, implementation in wave model, and evaluation. *Journal of Geophysical Research: Oceans*, 123(7):4779–4805, 2018.
[2] T. D. Williams, L. G. Bennetts, V. A. Squire, D. Dumont, and L. Bertino. Wave-ice interactions in the marginal ice zone. Part 1: Theoretical foundations. *Ocean Modelling*, 71:81–91, 2013.

---

## Author Response (AR1)

Dear Reviewer 1,

Thank you for taking the time and effort to read our manuscript and provide feedback. We found your comments very helpful and believe it will improve our manuscript. We have presented our response to the comments below in **blue and bold** and have indicated the changes to the manuscript in *italic*.
* * *
The fundamental difficulty in this from a theoretical point of view is that the sea state is random with a range of periods. It is therefore difficult to assign to any break-up event a single value for $\lambda$ unless it is for a wave tank experiment. This point should be discussed.

> **We agree with the reviewer that it is difficult to assign a specific value of $\lambda$ to the break-up events observed in this study and those reported in literature given that a wave field typically consists of a range of wave periods rather than a single period. Nevertheless, waves in sea ice are generally well-sorted due to the overwhelming dissipation of short waves a short way into the ice pack, such that the wave field is often narrow-banded and thus the peak wave length provides a reasonable choice to define the break-up. To attend the reader to this issue, we have extended the brief existing discussion (line 327, revised manuscript) on this topic with the following:**
>
> *"This emphasizes the difficulty in assigning a single characteristic wave length to a break up event for a wave field that is inherently random and consists of a range of length scales. Nevertheless, as short waves dissipate rapidly near the ice edge, the spectrum is often narrow-banded and thus the peak period is likely to be the most representative scale to characterize a break-up event."*

The equations for Young's modulus etc. are essential and summarise literature which is not well known. Multiple authors, mostly those associate with Squire, have used 6GPa for the Youngs modulus, which is an overestimate. However, it should be explained clearly what the units in the formulae are, and the units should be made consistent is possible (e.g. the units of brine volume).

> **This is a very valid point. We have made the units of $v_b$ and $S_{ice}$ consistent throughout the manuscript (fraction instead of ppt), see Eqs. 5-9 in revised manuscript.**

The breaking model, in fact, contains two contradictions/paradoxes. One is that in the limit of small thickness ice is unbreakable, and the other is that short-wavelength waves will break any ice. The second point seems to have been missed by the authors. However, the model assumes that the ice is moving compliantly with the sea surface and the wavelengths are so long that the sea ice can be modelled as a negligible surface. Some discussion of this point and the regime in which it is valid would be useful.

> **This is well noted by the reviewer. Indeed, our current definition of $I_{br}$ suggests that capillary waves, for example, would be able to break meters thick sea ice which is, of course, physically near impossible (aside from the fact that short waves won't penetrate far into the ice cover as they fully dissipate/scatter near the ice edge).**
> **We forgot to specify that Eq. 2 assumes that the ice sheet is thin compared to the wave length (i.e. $h/\lambda \ll 1$) and thus the break-up parameter $I_{br}$ cannot be applied to relatively short waves (i.e. $h/\lambda \gg 1$) as the ice is simply too 'heavy' to be impacted by short waves (and thus the ice will not move compliantly with the ice).**

In what range of $h/\lambda$ is the proposed threshold of $I_{br}$ valid? We currently have insufficient data to determine this, but the data (see Figure 8) suggest that at least up to $h/\lambda \approx 0.02$ the assumption seems to be valid. More data at higher values of $h/\lambda \approx 0.02$ are required to confirm a more definite regime.

We have added the 'thin plate' assumption to the Introduction. Specifically, we replaced 'an elastic plate' (line 57) by '*a thin elastic plate*'.

We have added the following to the Discussion section (starting at line 351, revised manuscript):

*While the current definition of $I_{br}$ suggests that very short waves always break the ice, it is worth reiterating that the assumption underlying Eq. 2 is that the ice is considered to be thin with respect to the wave length (i.e., $h/\lambda \ll 1$) and elastic (i.e. Eq. 2), implying that the ice moves compliantly with the sea surface. Thus, the threshold of $I_{br}$ defined in this study does not necessarily hold for short waves or, strictly speaking, for $h/\lambda \gg 1$. While the exact range of $h/\lambda$ for which the observed threshold of $I_{br}$ is valid is uncertain, based on the observations presented here (Figure 8), it seems that it upholds for $h/\lambda < 0.02$. More observations are required to clarify its validity for $h/\lambda = O(0.1 - 1)$. We note that this is not necessarily a limitation of the parameterization of $I_{br}$ as short waves are, in general, attenuated rapidly when entering the ice cover due to wave energy dissipation and scattering.*

The literature review is mostly complete. However, the first coupled attenuation and breaking model appeared in Kohout AL, Meylan MH. An elastic plate model for wave attenuation and ice floe breaking in the marginal ice zone. Journal of Geophysical Research: Oceans. 2008 Sep;113(C9). The authors concluded that their attenuation model was failing because of the overprediction of the break-up.

We thank the reviewer for this reference. We have integrated this reference in the introduction of the revised manuscript.

Dear Reviewer 2,

Thank you for taking the time and effort to read our manuscript and provide feedback. We found your comments very helpful and believe it will improve our manuscript. We have presented our response to the comments below in **blue and bold** and have indicated the changes to the manuscript in *italic*.
* * *
p3: section about $I_{br}$ could be improved - I have never heard the term "similitude" or of the "Pi-theorem" before - can you think of a better name? Using $ka$ (steepness) and $kh$ instead of converting $k$ to $\lambda$ would get rid of many factors of $2\pi$, and $I_{br}$ could become a stress relative to the flexural strength or a strain relative to the breaking strain for a beam. (The critical value would be about $4\pi^2 \times 0.014 = 0.55$ I guess). Since the relationship looks like it could have some universality it is worth presenting it somewhat more intuitively.

> **We appreciate the suggestion of the reviewer to use the wave number instead of the wave length. Despite the more attractive threshold value 0.55 (i.e. O(1)), we believe that the wave length is more intuitive than the wave number as it is a more 'direct observable' length scale in contrast to its inverse value (i.e. the wave number).**
>
> **The Pi-theorem is a theorem in dimensional analysis and, at least in our experience, is commonly applied in the fields of physics and engineering. As this is the conventional term to describe the theorem, we decided to keep this term and have provide a reference to the original theorem:** *"Buckingham (1914)"* **(line 54, revised manuscript). To improve reading, we have replaced the term "similitude" with** *"similarity"* **(line 53 of revised manuscript).**

"sheet as an elastice plate" – "sheet as a thin elastic plate" (or maybe simply an elastic beam, since you are using the $\sigma = Y\epsilon$ relation below).

> **We thank the reviewer for noting this as it is an important point. We will edit the manuscript to have it read** *"sheet as a thin elastic plate"* **(line 57, revised manuscript).**

p16: "infinitely thin ice sheet becomes numerically unbreakable" -- the opposite problem is that the strain as $kh \rightarrow \infty$ (shorter waves/thicker ice) also becomes infinite. In that case including reflection by ice edges is one way to reduce the strain inside the ice [1, 2]. Using the ice wavelength instead of the open water one could also make a difference here too. For both points the ice sheet example of Cathles et al jumps to mind.

> **This is an excellent point mentioned by the reviewer and also related to the previous comment. Indeed, our current definition of $I_{br}$ suggests that capillary waves, for example, would be able to break meters thick sea ice which is, of course, physically near impossible (aside from the fact that short waves won't penetrate far into the ice cover as they fully dissipate/scatter near the ice edge).**
> **We forgot to specify that Eq. 2 assumes that the ice sheet is thin compared to the wave length (i.e. $h/\lambda \ll 1$) and thus the break-up parameter $I_{br}$ cannot be applied to relatively short waves (i.e. $h/\lambda \gg 1$) as the ice is simply too 'heavy' to be impacted by short waves (and thus the ice will not move compliantly with the ice).**
>
> **To discuss this point further, we have added the following to the manuscript in the Discussion section (starting at line 351, revised manuscript):**

*While the current definition of $I_{br}$ suggests that very short waves always break the ice, it is worth reiterating that the assumption underlying Eq. 2 is that the ice is considered to be thin with respect to the wave length (i.e., $h/\lambda \ll 1$) and elastic (i.e. Eq. 2), implying that the ice moves compliantly with the sea surface. Thus, the threshold of $I_{br}$ defined in this study does not necessarily hold for short waves or, strictly speaking, for $h/\lambda \gg 1$. While the exact range of $h/\lambda$ for which the observed threshold of $I_{br}$ is valid is uncertain, based on the observations presented here (Figure 8), it seems that it upholds for $h/\lambda < 0.02$. More observations are required to clarify its validity for $h/\lambda = O(0.1 - 1)$. We note that this is not necessarily a limitation of the parameterization of $I_{br}$ as short waves are, in general, attenuated rapidly when entering the ice cover due to wave energy dissipation and scattering.*

**List of changes to manuscript:**

**Added, line 21:** *"(e.g. Kohout and Meylan, 2008*)"
**Added, line 33:** "*Kohout and Meylan, 2008*"
**Replaced, line 53:** similitude by "*similarity*"
**Added, line 54:** *"(Buckingham, 1914)"*
**Added, line 57:** "*thin*"
**Added, line 67:** "*Kohout and Meylan, 2008*"
**Replaced, line 194:** ppt by "*fraction*"
**Added, line 208:** "*1000*" in Equation for conversion of units
**Added, lines 327-330:**

[revised manuscript text omitted]

---

## Editor Decision (ED1)

**REVIEW OF 'EXPERIMENTAL EVIDENCE FOR A UNIVERSAL THRESHOLD CHARACTERIZING WAVE-INDUCED SEA ICE BREAK-UP'**

VOERMANS ET AL

**1. General comments**

This is a very nice collection of observations and the main result of figure 8 is surprising, but convincing with quite a few events included that represent many different scales and seem to have been careful with uncertainties. I recommend publication with minor revisions. I have a small suggestion about the definition of the $I_{br}$ threshold to make it more intuitive.

**2. Specific comments**

(1) p3: section about $I_{br}$ could be improved - I have never heard the term "similitude" or of the "Pi-theorem" before - can you think of a better name? Using $ka$ (steepness) and $kh$ instead of converting $k$ to $\lambda$ would get rid of many factors of $2\pi$, and $I_{br}$ could become a stress relative to the flexural strength or a strain relative to the breaking strain for a beam. (The critical value would be about $4\pi^2 \times 0.014 = .55$ I guess). Since the relationship looks like it could have some universality it is worth presenting it somewhat more intuitively.

(2) "sheet as an elastice plate" — "sheet as a thin elastic plate" (or maybe simply an elastic beam, since you are using the $\sigma = Y\varepsilon$ relation below).

(3) p16: "infinitely thin ice sheet becomes numerically unbreakable" — the opposite problem is that the strain as $kh \to \infty$ (shorter waves/thicker ice) also becomes infinite. In that case including reflection by ice edges is one way to reduce the strain inside the ice [1, 2]. Using the ice wavelength instead of the open water one could also make a difference here too. For both points the ice sheet example of Cathles et al jumps to mind.

**References**

[1] Guillaume Boutin, Fabrice Ardhuin, Dany Dumont, Caroline Sévigny, Fanny Girard-Ardhuin, and Mickael Accensi. Floe size effect on wave-ice interactions: Possible effects, implementation in wave model, and evaluation. *Journal of Geophysical Research: Oceans*, 123(7):4779–4805, 2018.

[2] T. D. Williams, L. G. Bennetts, V. A. Squire, D. Dumont, and L. Bertino. Wave-ice interactions in the marginal ice zone. Part 1: Theoretical foundations. *Ocean Modelling*, 71:81–91, 2013.

[Figure]

This paper presents an analysis of recent field campaigns in which the wave conditions and simultaneous sea ice break up were measured. This data is compared with data from previous observations to determine a threshold for breaking. Such a threshold will be useful provided it is accurate. Even if it is not entirely accurate, I believe it will serve as a valuable benchmark for comparison. I am supportive of the publication.

The fundamental difficulty in this from a theoretical point of view is that the sea state is random with a range of periods. It is therefore difficult to assign to any break-up event a single value for $\lambda$ unless it is for a wave tank experiment. This point should be discussed.

The equations for Young's modulus etc. are essential and summarise literature which is not well known. Multiple authors, mostly those associate with Squire, have used 6GPa for the Youngs modulus, which is an overestimate. However, it should be explained clearly what the units in the formulae are, and the units should be made consistent is possible (e.g. the units of brine volume).

The breaking model, in fact, contains two contradictions/paradoxes. One is that in the limit of small thickness ice is unbreakable, and the other is that short-wavelength waves will break any ice. The second point seems to have been missed by the authors. However, the model assumes that the ice is moving compliantly with the sea surface and the wavelengths are so long that the sea ice can be modelled as a negligible surface. Some discussion of this point and the regime in which it is valid would be useful.

The literature review is mostly complete. However, the first coupled attenuation and breaking model appeared in Kohout AL, Meylan MH. An elastic plate model for wave attenuation and ice floe breaking in the marginal ice zone. *Journal of Geophysical Research: Oceans*. 2008 Sep;113(C9). The authors concluded that their attenuation model was failing because of the overprediction of the break-up.